# Prophages integrating into prophages: A mechanism to accumulate type III secretion effector genes and duplicate Shiga toxin-encoding prophages in *Escherichia coli*

Keiji Nakamura[1], Yoshitoshi Ogura[2], Yasuhiro Gotoh[1], Tetsuya Hayashi[1]*

1 Department of Bacteriology, Graduate School of Medical Sciences, Kyushu University, Fukuoka, Japan,
2 Division of Microbiology, Department of Infectious Medicine, Kurume University School of Medicine, Kurume, Japan

* thayash@bact.med.kyushu-u.ac.jp

**Data Availability Statement:** All relevant data are within the manuscript and its Supporting Information files.

## Abstract

Bacteriophages (or phages) play major roles in the evolution of bacterial pathogens via horizontal gene transfer. Multiple phages are often integrated in a host chromosome as prophages, not only carrying various novel virulence-related genetic determinants into host bacteria but also providing various possibilities for prophage-prophage interactions in bacterial cells. In particular, *Escherichia coli* strains such as Shiga toxin (Stx)-producing *E. coli* (STEC) and enteropathogenic *E. coli* (EPEC) strains have acquired more than 10 prophages (up to 21 prophages), many of which encode type III secretion system (T3SS) effector gene clusters. In these strains, some prophages are present at a single locus in tandem, which is usually interpreted as the integration of phages that use the same attachment (*att*) sequence. Here, we present phages integrating into T3SS effector gene cluster-associated loci in prophages, which are widely distributed in STEC and EPEC. Some of the phages integrated into prophages are Stx-encoding phages (Stx phages) and have induced the duplication of Stx phages in a single cell. The identified *attB* sequences in prophage genomes are apparently derived from host chromosomes. In addition, two or three different *attB* sequences are present in some prophages, which results in the generation of prophage clusters in various complex configurations. These phages integrating into prophages represent a medically and biologically important type of inter-phage interaction that promotes the accumulation of T3SS effector genes in STEC and EPEC, the duplication of Stx phages in STEC, and the conversion of EPEC to STEC and that may be distributed in other types of *E. coli* strains as well as other prophage-rich bacterial species.

## Author summary

Multiple prophages are often integrated in a bacterial host chromosome and some are present at a single locus in tandem. The most striking examples are Shiga toxin (Stx)-producing and enteropathogenic *Escherichia coli* (STEC and EPEC) strains, which usually contain more than 10 prophages (up to 21). Many of them encode a cluster of type III

**Funding:** This research was supported by AMED under Grant Number 20fk0108065h0803 to T.H., and a KAKENHI from the Japan Society for the Promotion of Science (18K07116) to K.N. The funders had no role in study design, data collection and analysis, decision to publish, or preparation of the manuscript.

**Competing interests:** The authors have declared that no competing interests exist.

secretion system (T3SS) effector genes, contributing the acquisition of a large number of effectors (>30) by STEC and EPEC. Here, we describe prophages integrating into T3SS effector gene cluster-associated loci in prophages, which are widely distributed in STEC and EPEC. Two or three different attachment sequences derived from host chromosomes are present in some prophages, generating prophage clusters in various complex configurations. Of note, some of such phages integrating into prophages are Stx-encoding phages (Stx phages) and have induced the duplication of Stx phages. Thus, "prophage-in-prophage" represents an important inter-phage interaction as they can promote not only the accumulation of T3SS effectors in STEC and EPEC but also the duplication of Stx phages and the conversion of EPEC to STEC.

## Introduction

Horizontal gene transfer (HGT) is an important mechanism for generating genetic and phenotypic variations in bacteria [1–3]. Phages are major players in HGT, and many temperate phages that confer virulence potential to host bacteria through the transfer of virulence-related genes have been identified [4]. Most temperate phages integrate their genomes into host chromosomes by site-specific recombination to become a part of the chromosomes as prophages and enter a lysogenic cycle. Recombination takes place between the homologous sequences of phage and host DNA (*attP* and *attB*, respectively) and is mediated by a phage-encoded integrase [5]. Many bacterial species/strains contain multiple prophages [6–8], providing various possibilities for prophage-prophage interactions [9,10]. In particular, *Escherichia coli* strains such as Shiga toxin (Stx)-producing *E. coli* (STEC) strains have acquired more than 10 prophages (up to 21) [11–14], and some of the prophages are located at the same loci in tandem.

STEC strains cause diarrhea and severe illnesses, such as hemorrhagic colitis (HC) and life-threatening hemolytic-uremic syndrome (HUS). Their key virulence factor is Stx. While there are two subtypes (Stx1 and Stx2) with several variants and STEC produces one or more Stx subtypes/variants [15–18], the known *stx* genes are all encoded by prophage genomes. In addition, typical STEC strains share the locus of enterocyte effacement (LEE) encoding T3SS with enteropathogenic *E. coli* (EPEC), and more than 30 effectors have been carried into STEC and EPEC by multiple prophages [19–22]. Thus, EPEC strains are generally regarded as progenitors of typical STEC strains. For example, O157:H7 STEC evolved from an ancestral EPEC O55:H7 through the phage-mediated acquisition of *stx* along with a serotype change [23,24].

In this study, we initially analyzed the two copies of an Stx2-encoding prophage (referred to as Stx phage) present in two loci of an STEC O145:H28, one of the major types of non-O157 STEC [25,26], and found that one of them is integrated into another prophage. We then identified its *attB* sequence. By subsequent analyses of prophages carrying similar *attB* sequences, we show that phage integration in prophage (referred to as prophage-in-prophage) is a genetic event widely occurring in STEC and EPEC and represents a mechanism underlying the evolution and diversification of these bacteria.

## Results

### Integration of inducible and packageable Stx2a phages into a prophage integrated into the *ompW* locus in STEC O145:H28

We previously identified 18 prophages in the finished genome of O145:H28 strain 112648 [27], including two Stx2a phages found at the *ompW* (P09) and *yecE* loci (P12). The two Stx2a phage genomes were identical in sequence; thus, they were considered duplicated prophages.

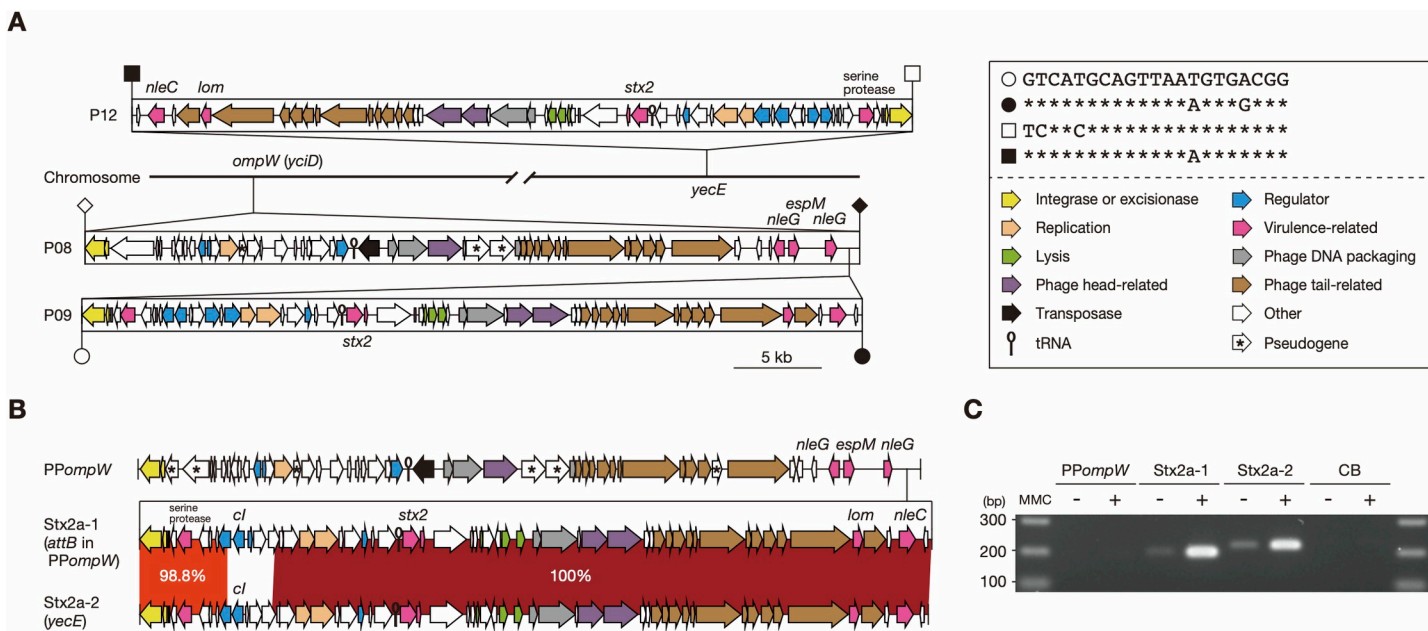

**Fig 1. Integration sites of the inducible and packageable duplicated Stx2a phages in two STEC O145:H28 strains.** (A) The duplicated Stx2a phages and their *att* sequences in strain 112648. The genome structures of three prophages (P08, P09, and P12) are drawn to scale. The *att* sites of each prophage are indicated by open (*attL*) or filled (*attR*) symbols (P08, rhombus; P09, circle; P12, square). The *att* sequences of the Stx2a phages (P09 and P12) are shown in the inset. (B) The genome structures of two Stx2a phages and a lambda-like phage integrated into *ompW* (PP*ompW*) in strain 12E129. Sequence homology between the two Stx2a phages is also shown, with their integration sites indicated in parentheses. Homologous regions are indicated by shading with different colors according to sequence identity. The integrase gene of PP*ompW* and the *cI* gene on each Stx2a phage were targeted by the PCR primers used in Fig 1C. (C) Detection of packaged DNA of the three prophages in the DNase-treated lysates of strain 12E129 with (+) or without (-) MMC treatment. The chromosome backbone (CB) region was amplified as a negative control.

As the lambda-like P08 prophage was also found at *ompW*, we initially thought that P08 and P09 had been integrated in tandem. However, by analyzing the potential *att* sequences of the three PPs, we found that, while P08 was integrated into the *ompW* gene with an *attL/R* sequence of 121 bp, P09 was integrated into the P08 genome with a 21-bp *attL/R* sequence which is similar to the *attL/R* of P12 in the *yecE* gene (Fig 1A). By analyzing the prophages at the *ompW* and *yecE* loci in O145:H28 strains, we identified another strain (12E129) that carries the same set of prophages: a lambda-like prophage at *ompW*, an Stx2a phage in the prophage at *ompW*, and another Stx2a phage at *yecE*. The potential *att* sequences of the three prophages were identical to those of the corresponding prophages in strain 112648 (S1 Fig). Although the genomes of the two Stx2a phages in strain 12E129 were also nearly identical, the left end regions including the *cI* gene significantly differed in sequence (Fig 1B). Therefore, there are two possibilities for the acquisition of two copies of Stx2a phage by this strain; the left end segment has been replaced by recombination after duplication or two Stx2a phages were infected independently. Hereafter, prophages integrated into the same locus are collectively referred to as PPxxx (where xxx denotes the integration locus), such as PP*ompW*.

To precisely determine the *att* sequences of each prophage, we amplified and sequenced the *attP*-flanking regions of excised and circularized genomes of these phages. Although the two Stx2a phages in strain 112648 were indistinguishable, those of strain 12E129 were distinguishable, allowing sequence determination of the *attP*-flanking regions of three prophages from mitomycin C (MMC)-treated cell lysates. This analysis confirmed that the predicted *att* sequences exactly represented those of the three prophages and revealed that their genomes were excised and circularized by MMC treatment (S1 Fig). The *att* sequences of P08 and P09/

P12 in strain 112648 were also confirmed using the same strategy. These results indicate that, in both strains, one of the two Stx2a phages has been integrated into PP*ompW*.

We further examined the packageability of these prophage genomes into phage particles by PCR analysis of DNase-treated culture supernatants of strain 12E129 with or without MMC treatment (Fig 1C). This analysis detected DNase-resistant genomic DNA of the two Stx2a phages, but did not that of PP*ompW*, indicating that these Stx2a phages were both packaged into the phage particles. In a similar analysis of strain 112648, the packaged genome of Stx2a phage (P09 and/or P12) was detected (S1D Fig). That of P08 (PP*ompW*) was also not detected. The deficiency of the two PP*ompW*s in genome packaging can be explained by the mutation in a gene for head formation (Fig 1A and 1B). These PP*ompW*s contained an insertion sequence (IS)-mediated deletion of the lysis gene cassette-encoding region and one or more additional pseudogenes, thus both are apparently defective.

## Dynamics of PP*ompW*s, phages integrated into PP*ompW*, and PP*yecE*s in STEC O145:H28

To investigate the distribution of PP*ompW*s and the *attB* sequences found in two PP*ompW*s among O145:H28 strains, we selected 64 genomes from 239 strains analyzed in our previous study (S1 Table) [27]. This set comprised 8 finished and 56 draft genomes and encompassed seven of the eight clades previously identified in the major lineage (sequence type (ST) 32) and a minor lineage (ST137/6130) of O145:H28, thus largely representing the overall phylogeny of O145:H28 as shown by a whole genome-based maximum likelihood (ML) tree (Fig 2).

PP*ompW*s were present in all 64 strains analyzed, including the two aforementioned strains. All-to-all sequence comparison of the PP*ompW*s from eight finished genomes and 12 PP*ompW*s sequenced in this study revealed that the PP*ompW* genomes were highly conserved with average nucleotide identity (ANI) values of >97.0%, although sequence diversification and segment replacement, probably by recombination, were detected in some parts of several PP*ompW*s (S2A Fig). Further analysis of the 20 PP*ompW*s revealed that all contained the 21-bp *att* sequence (Fig 2), with one exception where the 21-bp *attB*-containing region had been lost by IS insertion. These results indicate that a PP*ompW* containing the 21-bp *attB* sequence was acquired by an ancestral strain and has been stably maintained in O145:H28. It should be noted that, as seen in the two aforementioned strains, all PP*ompW*s analyzed in this study contained multiple gene inactivations and deletions, and thus appear to be defective.

Examination of the *attB* in PP*ompW* and *yecE* loci in the 64 strains revealed that prophages are integrated into the two loci in 14 and 21 strains, respectively, with marked variation in the prophage content between strains (Fig 2). At the *attB* in PP*ompW*, Stx2a phages were present in 10 strains and non-Stx phages in four strains (all belonging to ST32 clade H). More variable prophages were found at *yecE*: Stx1a phages in 11 strains, Stx2a phages in eight strains, Stx2d phage in one strain, and non-Stx phages in two strains. Two aforementioned strains (112648 and 12E129) carrying two Stx2a phages belonged to different ST32 clades, indicating that their acquisition of two copies of Stx2a phages occurred independently.

All-to-all sequence comparison of 27 prophage genomes integrated into the *attB* in PP*ompW* (n = 8; all were Stx2a phages) or *yecE* (n = 19; 9 Stx1 phages, 8 Stx2a phages, one Stx2d phage, and one non-Stx phage) locus revealed that the Stx1a phage genomes were relatively well conserved (ANI: >98.1%), with several regions with sequence divergence probably introduced by recombination (S2B Fig). In contrast, the Stx2a phage genomes were highly variable except for those in the ST32 clade A/B/C strains. Interestingly, although the Stx2a phage of strain RM9872 (clade C) was integrated into *yecE*, this phage was similar to the Stx2a phage at the *attB* in PP*ompW* in clade A/B strains (ANI: >98.1%). Considering the high conservation

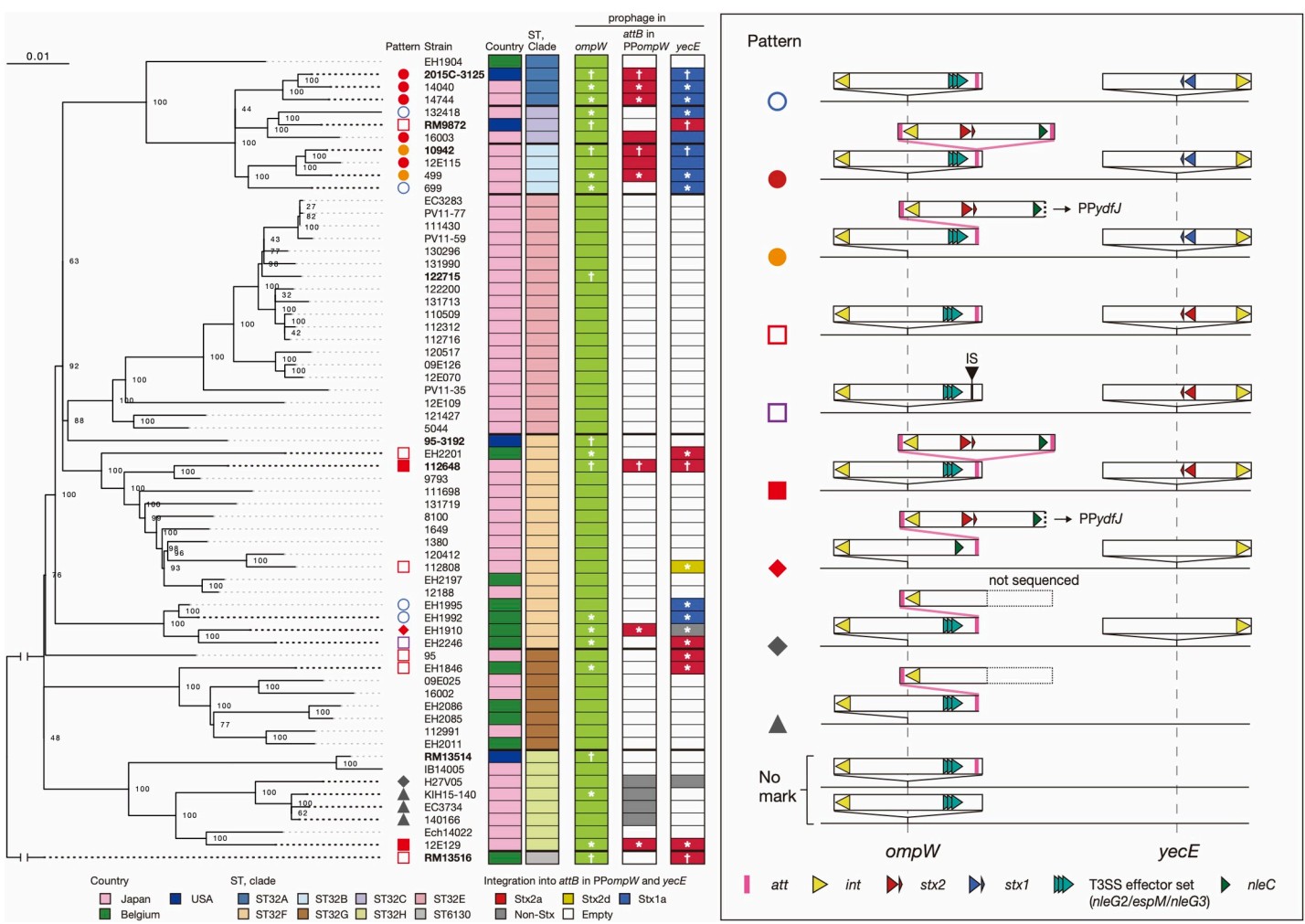

**Fig 2. Variation in the prophage content at the *ompW*, *attB* in PP*ompW*, and *yecE* loci in STEC O145:H28.** In the left panel, an ML tree of 64 O145:H28 strains is shown. Completely sequenced strains are indicated in bold (plasmids were not finished for strain 2015C-3125). The tree was constructed based on the recombination-free SNPs (3,277 sites) identified on the conserved chromosome backbone (3,961,936 bp in total length) by RAxML using the GTR gamma substitution model [43]. The reliabilities of the tree's internal branches were assessed using bootstrapping with 1,000 pseudoreplicates. Along with the tree, the geographic and ST/clade information of strains, the presence or absence of prophages at three loci (*ompW*, *attB* in PP*ompW*, and *yecE*) and the types of prophages at the *attB* in PP*ompW* and *yecE* loci are shown. Prophages sequenced in this study and those in the finished genomes are indicated by asterisks and daggers, respectively. Note that the *attB* in PP*ompW* sequence is missing from the PP*ompW*s of strains EH2246 and 12E109; a deletion in the latter stain was detected in its draft genome assembly. The bar indicates the mean number of nucleotide substitutions per site. In the right panel, the patterns of the prophage content at the three loci are schematically presented. Strains showing each pattern are also indicated in the left panel by diagrams. For more details about the T3SS effector set, see Fig 3 and main text. Note that we detected recombination between the Stx2a phage at *attB* in PP*ompW* and a prophage present at the *ydfJ* locus that induced a large chromosome inversion in strains 10942 and 499. In strain EH1910, an inversion appeared to have occurred by the recombination between PP*ompW* and a prophage at *ydfJ*.

of Stx1a phages at the *yecE* locus in these clades, it is likely that the Stx1a phage at *yecE* has been replaced by the Stx2a phage originally integrated into the PP*ompW* in strain RM9872.

## Wide distribution of PP*ompW*s and the 21-bp *attB* sequence in *E. coli*

We next examined the distribution of PP*ompW*s and the 21-bp *attB* sequence (or sequences similar to it) in the entire *E. coli* lineage by searching for them in all publicly available complete *E. coli* genomes (n = 767). Although this set was biased to STEC/EPEC strains (n = 252; S2 Table), particularly STEC O157 (n = 91; S3 Table), we used this set because detailed and accurate analyses of prophage configurations and sequences were possible only in complete

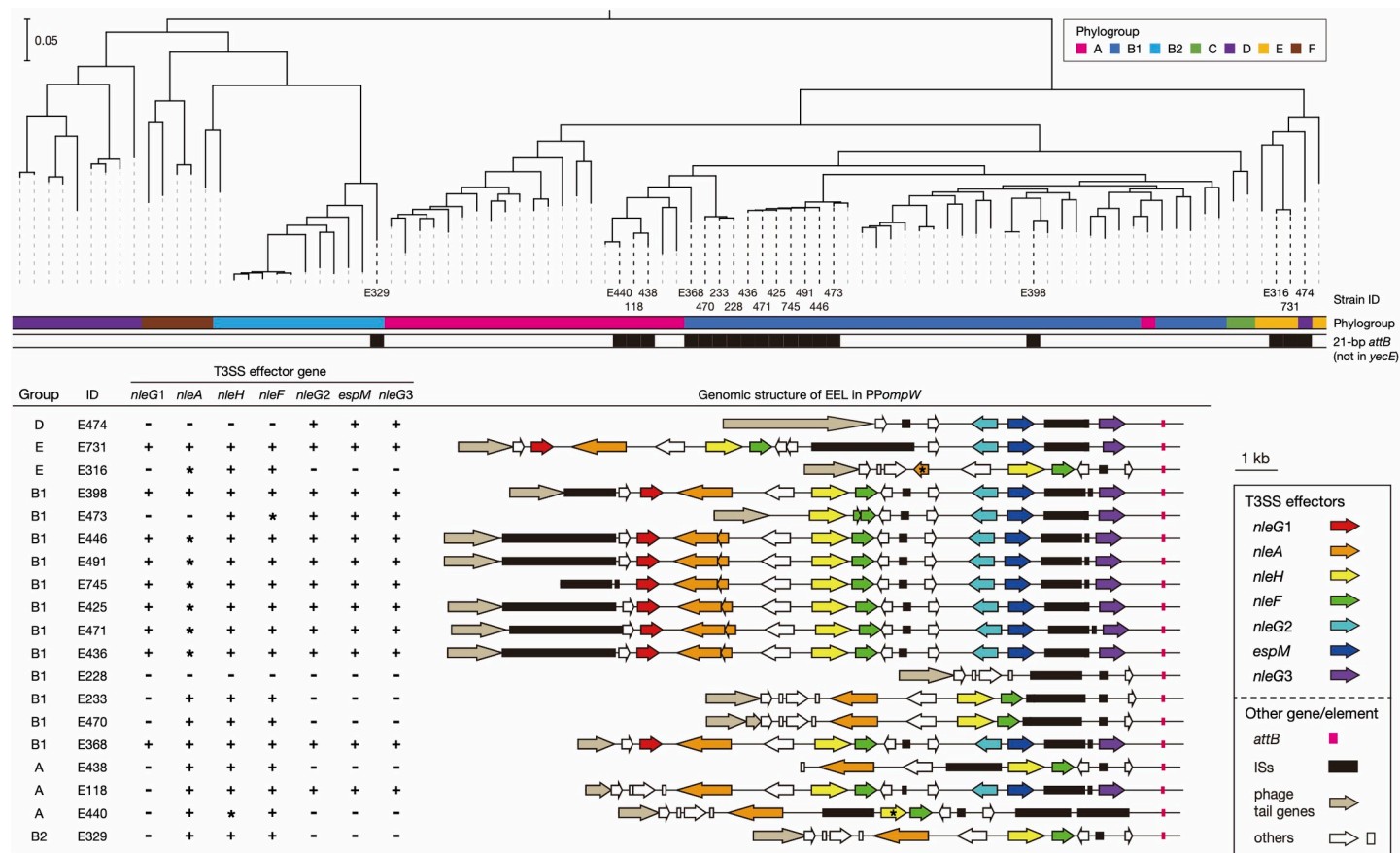

**Fig 3. Phylogenetic positions of *E. coli* strains carrying PP*ompW* and the genome structures of their EELs associated with 21-bp *attB* sequence.** In the upper panel, an ML tree of 92 complete genomes of *E. coli* strains that carry PP*ompW* is shown. The tree was constructed based on 109,927 SNP sites in 2,642 core genes and rooted by cryptic *Escherichia* clade I strain TW15838 (No. AEKA01000000) used as an outlier. Along with the tree, strain IDs used in this paper (see S5 Table for more details), phylogroups, and the presence (colored) or absence (open) of 21-bp *attB* sequence in each strain are indicated. The bar indicates the mean number of nucleotide substitutions per site. In the lower panel, the repertoires of T3SS effector genes that were encoded by the effector exchangeable loci (EELs) in the PP*ompW*s containing the 21-bp *attB* sequence are shown. The genomic structures of EELs are drawn to scale. All effector genes were aligned using BLASTN, and orthologous genes (sequence identity; >90%, coverage; >90%) are indicated by the same color. Genes with over 90% identity but less than 90% coverage and those containing indels and nonsense mutations in the sequence alignment to intact genes are indicated by asterisks.

genomes. PP*ompW* was found in 44% of the *E. coli* strains examined (338 strains of 92 serotypes; all but O145:H28 and O26:H11 comprised a single ST). Phylogenetic analysis of the *E. coli* strains representing each of the 92 serotypes showed that PP*ompW*s are widely distributed in *E. coli* (Fig 3). In contrast, after excluding the 21-bp *attB* sequence in *yecE*, 21-bp sequences identical to the *attB* in the PP*ompW* of O145:H28 strain 122715 (S1C Fig) or with a 1-base mismatch (hereafter, collectively referred to as 21-bp sequences) were detected in 150 strains of 20 serotypes belonging to five different *E. coli* phylogroups (Fig 3 and S3 Table). In 145 of the 150 strains, the 21-bp sequence was present in PP*ompW*s, but 28 of the 145 strains (all were serotype O157:H7) contained an additional 21-bp sequence in the second PP*ompW* located in tandem (n = 4), a prophage cluster at *mlrA* (n = 1), or prophage clusters at *ydfJ* (n = 23). Among the remaining five strains, one atypical O157:H7 strain (PV15-279) contained a 21-bp sequence in a prophage cluster at *ydfJ* and four strains (all were serotype O177:H25) contained one or two 21-bp sequences in a prophage or prophage cluster at *ydfJ*. (Note that PP*ompW* was not present in these O177:H25 strains.)

By examining the 145 strains containing the 21-bp sequence in PP*ompW*, we identified three non-O145:H28 strains carrying prophages integrated in PP*ompW*: one O157:H7 strain and two O145:H25 strains (S4 Table). Moreover, similar to the two aforementioned O145:H28 strains, the acquisition of two copies of Stx2 phage (one in the *attB* in PP*ompW* and the others in the *yecE* loci) occurred in two of the three strains (Stx2d phage in O157:H7 strain 28RC1 and Stx2a phage in O145:H25 strain CFSAN004176). These Stx2 phages appear to have been duplicated in each strain because the two Stx2 phages in each strain were nearly identical across the entire genome (ANI: >99.8%; S3 Fig), although one Stx2a phage in the O145:H25 strain contained a large genomic deletion and its *stx2A* gene was inactivated by multiple insertions and deletions [14].

## Close association of the 21-bp *attB* sequence with the prophage regions encoding T3SS effector genes

Comparison of the PP*ompW* genomes containing the 21-bp *attB* sequence (S4 Fig) revealed that while the early regions were relatively well conserved, the late regions were highly variable. In particular, the PP*ompW* genomes of phylogroup A strains have been highly degraded by deletions. Other PP*ompW* genomes also contained IS insertions and multiple gene inactivation and deletion, and all PP*ompW*s appear to be defective, as seen in O145:H28 strains. In fact, we detected no packaged DNA of the PP*ompW* of strain E2348/69, which contained a relatively well conserved gene set (E329 in Figs 3 and S4). However, multiple T3SS effector genes are present just upstream of the 21-bp *attB* sequence in all PP*ompW*s including those of O145:H28 strains. Only exception was that in an O182:H25 strain, from which effector genes have apparently been deleted (Figs 3 and S4). Thus, the 21-bp *attB* sequence is closely linked to the T3SS effector-encoding locus located at the very end of PP*ompW* genomes. Such regions of lambda-like phages encoding various T3SS effector genes are called exchangeable effector loci or EELs [20]. All PP*ompW*s containing the 21-bp *attB* sequence were also lambda-like phages.

By analyzing T3SS effector genes in the EELs in the PP*ompW* genomes, we identified seven effector genes belonging to the *nleA*, *nleH*, *nleF*, and *espM* families and three *nleG* subfamilies (*G*1-3) (Fig 3). Although there were variations in the effector gene repertoire between PP*ompW*s and gene inactivation due to various types of mutations (mostly deletions) was detected in several PP*ompW*s, a similar set of effector genes was found at the PP*ompW* EELs. As one or more IS elements were present at all EELs, the variation in effector gene repertoire was probably generated by IS insertion-associated events. The conservation patterns of effector genes among the 19 EEL-positive PP*ompW* genomes suggest that the EELs of O157:H7 strain Sakai (phylogroup E, E731 in Fig 3) and EPEC O76:H7 strain FORC_042 (phylogroup B1, E398 in Fig 3) represent the ancestral structure encoding seven effector genes.

It should be noted that, in all 150 strains that were found to contain the 21-bp *attB* sequence, the sequence was associated with these seven effector genes or a single *nleG* variant (see S5 Table for the details). These strains possessed the *eae* gene, a marker gene of the LEE (S3 Table and S5 Table for more details on each strain) with no exception, indicating that they are all EPEC or typical (LEE-positive) STEC.

## Prophage clusters that contained prophages carrying the 21-bp *attB* sequence and identification of additional *attB* sites in prophage genomes

Among the aforementioned 28 O157:H7 strains that contained two 21-bp *attB* sequences, four strains contained these sequences in the EEL-associated region of two PP*ompW* genomes integrated in tandem (Fig 4A and type a1 in S5 Fig). In these strains (as represented by FRIK2069 in Fig 4A), while one of the EELs encoded an effector gene set similar to that of other PP*ompW*

EELs, the other encoded an *nleG* variant different from the three *nleG* subfamilies at other PP*ompW* EELs.

In other 24 O157:H7 strains, one 21-bp sequence was present in PP*ompW*, and the other was present in prophage clusters comprising two to four prophages. In one of the 24 strains (FRIK944; Fig 4B and type b in S5 Fig), the prophage cluster was present at *mlrA* (synonyms: *yehV*) and comprised two prophages, an Stx1 phage and a lambda-like phage. By analyzing the *attL/R* sites of each prophage, we found that while Stx1 phage is integrated into *mlrA* [9], the lambda-like phage has been integrated into the Stx1 phage, using the 96-bp *att* sequence (see S6 Fig for the sequence) which is associated with an EEL similar to PP*ompW* EELs. The lambda-like phage also contained an *nleG* variant, but the 21-bp sequence was present between *attL* and the integrase gene and was not associated with the *nleG* variant. Hereafter, we referred to the 96-bp *att* sequence as *attB*-in-PP_2 and the 21-bp *attB* sequence which first identified in PP*ompW*s as *attB*-in-PP_1. Intriguingly, between the *attB*-in-PP_1 and the integrase gene of the lambda-like phage, the 121-bp *attB* sequence for PP*ompW*s was present. Although phage integration into the 121-bp sequence in prophage genomes has yet to be identified, this sequence can serve as a potential *attB* site in prophage genomes. We therefore refer to it as *attB*-in-PP_3.

In the remaining 23 O157:H7 strains, prophage clusters comprising two to four prophages were present at *ydfJ* (as represented by that of strain PV15-279 in Fig 4C; see types c2 and c 3 in S5 Fig for those of other strains). In these strains, one or two lambda-like phages, which carry EELs similar to the PP*ompW* EELs or encode multiple *nleG* variants, were integrated into *ydfJ* (see S7 Fig for the *attB* sequences). The former type of EEL was associated with *attB*-in-PP_2, into which another lambda-like phage was integrated. The phages integrated into PP*ydfJ* contained the *attB*-in-PP_1 and *attB*-in-PP_3 sequences downstream of the integrase gene and encoded *nleG* variants at the opposite end. This organization is similar to that of the above-mentioned phage integrated into PP*mlrA* (Fig 4B). Moreover, in strain PV15-279 shown in Fig 4C (an atypical O157:H7 strain [28]), an Stx2a phage was integrated into the *attB*-in-PP_1 of the prophage integrated into PP*ydfJ*.

Among the four aforementioned O177:H25 strains that contained one or two *attB*-in-PP_1 sequences, a similar but slightly different pattern of prophage integration into prophage genomes was observed (Fig 4D and types d1 and d2 in S5 Fig). In these strains, the *attB*-in-PP_1 sequence was found in a region that comprised two highly degraded prophages integrated in tandem between the *trg* and *rspB* genes. EELs similar to the PP*ompW* EELs, *attB*-in-PP_2 and *attB*-in-PP_1 were found in this order, and a lambda-like phage was integrated into *attB*-in-PP_2. Moreover, the lambda-like phages integrated into *attB*-in-PP_2 contained the *attB*-in-PP_1 and *attB*-in-PP_3 sequences and multiple *nleG* variants, similar to the phages integrated into PP*mlrA* or PP*ydfJ*s (Fig 4D). This finding indicates that the distribution of these three *attB* sequences in prophage genomes is not limited to O157:H7 strains.

It should be noted that, similar to PP*ompW*s, many of the prophages in the prophage clusters described in this section, including all prophages which were found to be integrated by other phages, are apparently defective due to small or large genome deletions and multiple gene inactivation (S5 Fig).

## Origins of *attB* sequences in prophages

Finally, to explore the origins of these *attB* sequences in prophages, we compared their flanking sequences with *E. coli* chromosome sequences. The *attB*-in-PP_1-flanking sequences in PP*ompW*s and other prophages (all are integrated into prophages as shown in S5 Fig) were highly conserved (S8 Fig), implying that these sequences have a common origin. Moreover,

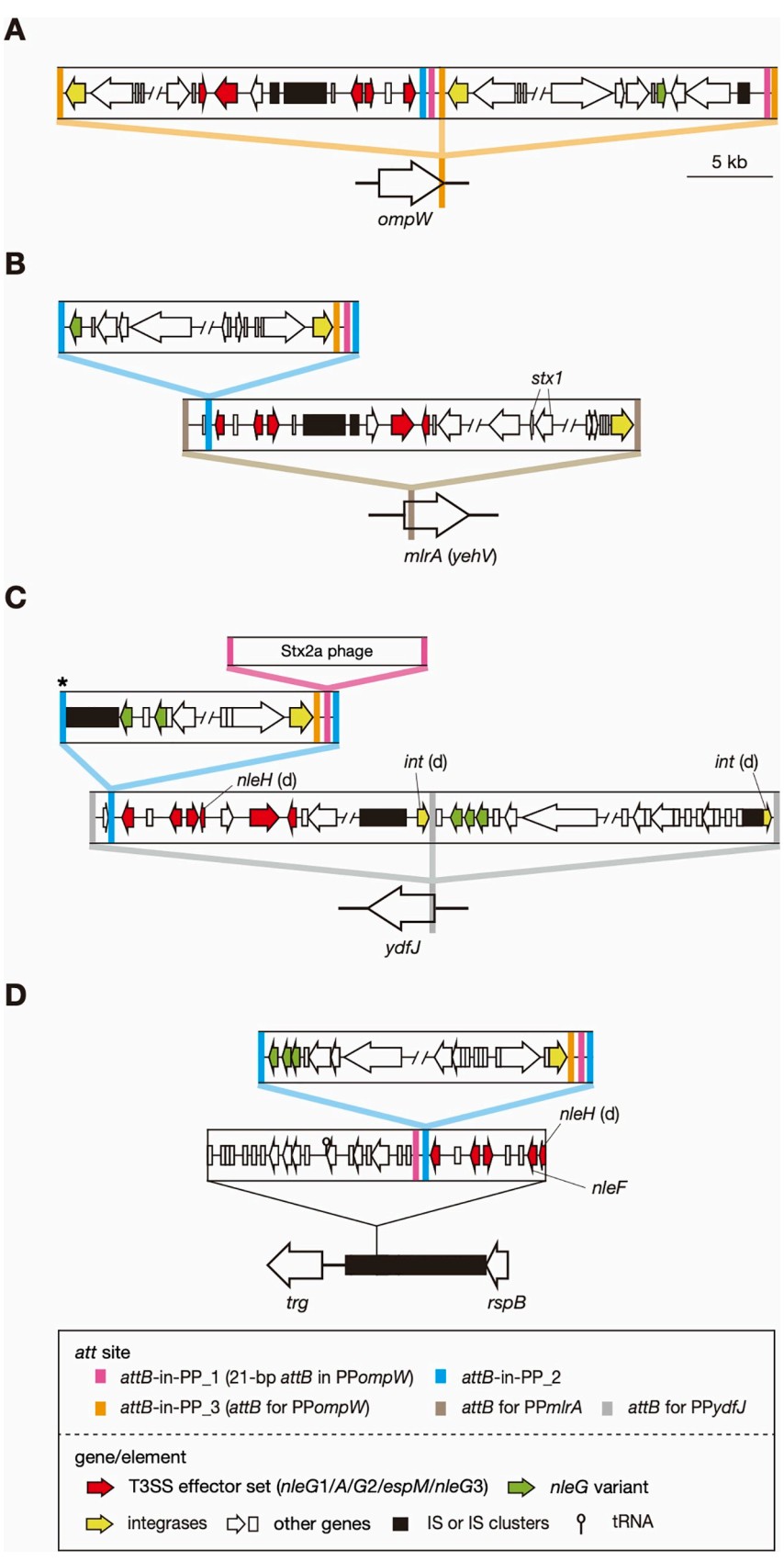

**Fig 4. Prophage clusters that contained prophage carrying potential *att* sequences in O157:H7 and O177:H25 strains.** The genomic structures of three representative prophage clusters of the 33 clusters found in O157:H7 and that of O177:H25 strains are shown (A, strain FRIK2069; B, strain FRIK944; C, atypical O157:H7 strain PV15-279; D, O177:H25 strain SMN152S1). The identified *attB* sequences, coding sequences (CDSs) (including pseudogenes), and ISs in each prophage are indicated. T3SS effector genes found in the PP*ompW* EELs (Fig 3) and other effector genes (*nleG* variants) are distinguished by different colors. In panel C, the *attB* sequence indicated by an asterisk is truncated by an IS insertion, and integration of an Stx2a phage into the *attB*-in-PP_1 site is schematically presented. The integrase (*int*) genes and the *nleH* genes that have been degraded are indicated by (d). The genome structures of all prophage clusters identified in this analysis are illustrated in S5 Fig.

the 100-bp sequences including the 21-bp *attB*-in-PP_1 sequence showed a notable similarity (87% identity) to the corresponding *yecE* region (Fig 5; see S8 Fig for sequence alignment), suggesting that the *attB*-in-PP_1 and its flanking sequence originated from the *yecE* locus.

Sequence similarity was also detected between the 96-bp *attB*-in-PP_2 sequence in the PP*ompW*s and the *ykgJ*/*ecpE* intergenic region of the *E. coli* chromosome (about 81% identity) (Fig 5; see S9 Fig for more details including sequence alignment). As the homologous sequence extended to 125 bp in PP*mlrA* and PP*ydfJ* (S9 Fig), we performed an additional search of *E. coli* complete genomes and identified seven *attB*-in-PP_2-containing prophages, although this search was limited to six STEC genomes, in which their prophages were fully annotated. The prophages identified in this search included the Stx1a phage (Sp15) at *mlrA* of O157:H7 strain Sakai [11], the aforementioned duplicated Stx2a phages of O145:H28 strain 112648, duplicated Stx2a phages of the atypical O157:H7 strain PV15-279 (one in PP*ompW* and the other in *yecE*; carrying an *nleC* effector gene), and two non-Stx phages in O26:H11 and O111:H8 STEC strains [12] (at *ydfJ* and *ssrA*, respectively; the former carries an *nleC* effector gene). In these seven prophages, homologous sequences further extended to 309 bp with 84% identity (Figs 5 and S9). Contrary to the observation for *attB*-in-PP_1 and its flanking sequences, there was notable diversity in the *attB*-in-PP_2 sequence (23/96 polymorphic sites) between the

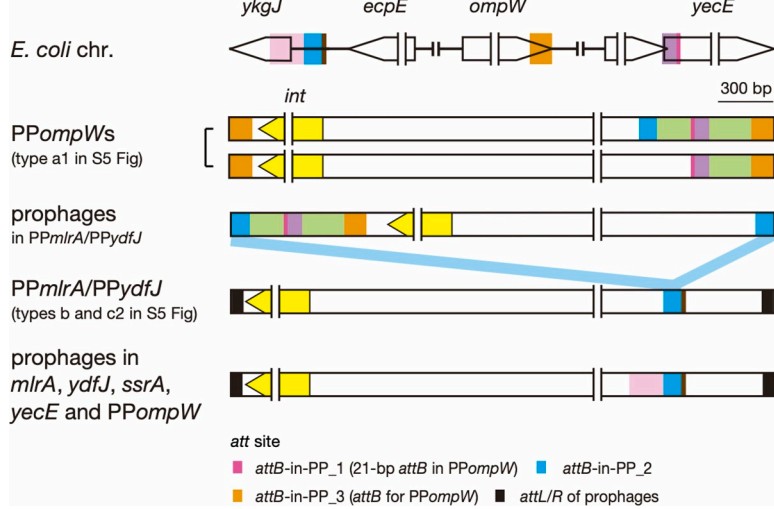

**Fig 5. Locations of the *attB*-in-PP sequences in prophages and the prophage genome regions homologous to *E. coli* chromosome regions.** Three loci in the *E. coli* chromosome showing sequence homology to three identified *attB*-in-PP sequences and their flanking sequences are shown at the top. The left- and right-end regions of representative prophages that contained the *att*-in-PP sequences are shown below. Homologous sequences are indicated by the same color. The color used for each *attB*-in-PP sequence is the same as that used in Figs 4 and S5. See these figures for the details of "PP*ompW*s" and "prophages in PP*mlrA*/PP*ydfJ*" and S9 Fig for information on the prophages in *mlrA*, *ydfJ*, *ssrA*, and *yecE* and PP*ompW*. Alignments of the *attB*-in-PP_1 and *attB*-in-PP_2 sequences and their flanking sequences with corresponding chromosome sequences are shown in S8 and S9 Figs, respectively.

PP*ompW*s, PP*mlrA* in strain FRIK944, PP*ydfJ* in strain 141 and the other seven prophages (S9 Fig). These findings indicate that the *attB*-in-PP_2 and its flanking sequences originated from the *ykgJ*/*ecpE* intergenic region on the chromosomes of *E. coli* or its close relatives, but acquisition of the sequences by phages might have occurred multiple times.

The 121-bp *attB*-in-PP_3 sequence which was found in many of the phages integrated in prophages identified in this study (Figs 4 and 5, see S5 Fig for more details) showed 81% identity to the *E. coli ompW*, suggesting that its possible origin is also the chromosome of *E. coli* or its close relatives. Interestingly, PP*ompW*s and many other phages integrated in prophages contained two or three *attB*-in-PP sequences in the same order. The sequences between the *attB*-in-PP sequences (indicated by green in Fig 5) were also conserved (up to a 5-single nucleotide polymorphism (SNP) difference); however, the location of the *attB*-in-PP set in PP*ompW*s was different from that of other phages integrated in prophages. This finding suggests that the region encompassing three (or two) *attB*-in-PP sequences was once acquired by either type of phage and spread to the other by recombination between prophages or some other mechanisms.

## Discussion

As summarized in Fig 6, we identified various phage integration patterns in STEC and EPEC strains, including phage integration into prophages. Most temperate phages are integrated into host genomes by integrase-mediated recombination between *attP* and *attB*. Tandem integration can occur if the two phages share the same *attB* site. In contrast to this traditional view of the mechanism for generating tandem prophages, this study identified many prophages that contain *attB* sequences, which allow another phage to be integrated into their genomes, forming a prophage-in-prophage configuration. The combination of the two integration mechanisms generates more complex prophage clusters in host genomes (Fig 6; combination of tandem prophages and prophage-in-prophage). Frequent colocalization of multiple *attB* sequences in prophages potentially generates much more variation than detected in this study. These *attB* sequences probably originated from the host chromosome, providing more opportunities for lysogenization to incoming phages and allowing the duplication of prophages encoding medically or biologically important genes, such as *stx*.

Interactions between integrating prophages and their "host" prophages are an important issue to better understand the processes of prophage-in-prophage formation. In this regard, a notable finding is that all host prophages identified in this study were apparently defective. This finding suggests that *attB* sequences in defective prophages are favorable sites for phage integration, which represents a hidden biological role of defective phages. The mechanism(s) of incorporating *attB* sequences into phage genomes from host chromosomes is another important issue. Although we were unable to address the molecular mechanism(s) in this study, intra-chromosomal recombination between prophages might be involved in this process. Such recombination, particularly those inducing symmetric chromosome inversions in respect to the *ori-ter* axis, could also contribute to the formation and variation of complex prophage clusters and the spread of intra-prophage *attB* sequences. In fact, we observed recombination and chromosome inversion between prophages/prophage clusters at *ompW* and *ydfJ* in several strains (Figs 2 and S5).

Notably, most *attB* sequences in prophages identified are linked to EELs that encode multiple effector genes for the LEE-encoded T3SS, and prophages integrated into the *attB* in prophages often carry *nleC* and *nleG* family effector genes. Thus, the prophage-in-prophage system has promoted the accumulation and variation of effector genes in EPEC and STEC strains [21,29], although how it affects the virulence potential of these strains is currently

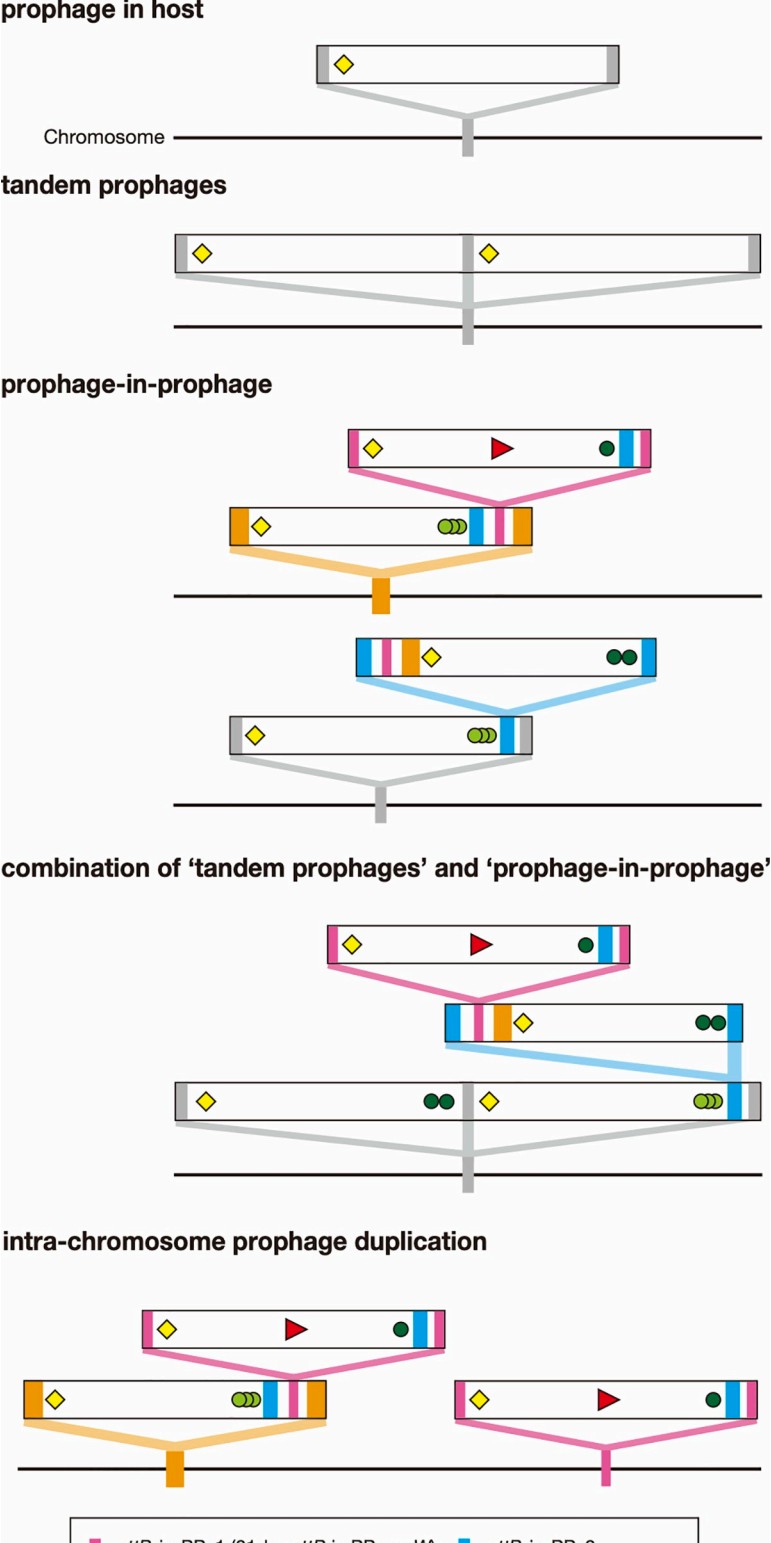

**Fig 6. Summary of the variable phage integration patterns found in this study.**

unknown and requires further analyses [22,30]. It is also noteworthy that a significant portion of the phages integrated in *attB*-in-PP_1 (13/18) encoded *stx* genes, indicating that the *attB*-in-PP_1 sequence can contribute to the acquisition of *stx* genes and thus the conversion of EPEC to typical STEC, even if the *yecE* locus, the origin of *attB*-in-PP_1 and one of the integration hot spots of Stx phage [12,31–33], has been occupied by another phage.

In conclusion, the findings obtained here highlight that phage integration systems are much more complicated than previously recognized and provide additional insights into the evolution of EPEC and STEC and their genetic diversity. It may be also possible to find similar prophage integration patterns in other types of *E. coli* and other prophage-rich species if prophage clusters are carefully investigated. Similar integration systems could also be found for genetic elements utilizing integrase-mediated integration mechanisms, such as integrative and conjugative elements (ICEs) [34].

## Material and methods

### Bacterial strains

The 64 O145:H28 strains analyzed in this study are listed in S1 Table. Of these, 59 were from our laboratory stock, which were genome-sequenced in our previous study [27], and 5 were completely genome-sequenced stains (the plasmid genome was not finished in strain 2015C-3125), the genome sequences of which were downloaded from the NCBI database. To construct the completely genome-sequenced *E. coli* strain set, a total of 875 complete genomes were downloaded from the database (accessed on the 20th of July 2019). After excluding laboratory, commercial and re-sequenced strains and substrains, the 767 strains listed in S5 Table were used for analysis. Annotation was carried out using the DDBJ Fast Annotation and Submission Tool (DFAST) [35], if necessary.

### Extraction of total cellular and phage DNA

Bacterial cells were grown overnight to the stationary phase at 37°C in lysogeny broth (LB) medium. For prophage induction, cells were grown to the late log phase (0.7–0.9 $OD_{600}$), and MMC was added to the culture to a final concentration of 1 μg/ml. After a 3-hr incubation, aliquots of the culture were isolated, and the cells were collected by centrifugation. Total cellular DNA was extracted from the cells using the alkaline-boiling method and used for PCR analyses. Phage particles were isolated from the culture supernatant after a 3-hr incubation with MMC. The culture was first treated with chloroform, and bacterial cell debris was removed by centrifugation. The supernatant was filtered through a 0.2-μm-pore-size filter (Millipore) and incubated with DNase I (final concentration: 400 U/ml, TaKaRa) and RNase A (50 μg/ml, Sigma) at 37°C for 1 hr. After inactivating DNase I by incubation at 75°C for 10 min and adding EDTA (5 mM, Nacalai Tesque), the sample was treated with proteinase K (100 μg/ml; Wako) and used as packaged phage DNA. Total cellular DNA and packaged phage DNA from MMC-untreated cultures were prepared with the same protocol. The primers used in these analyses are listed in S6 Table.

### Analyses of phage integration and sequencing of prophage genomes

Phage integration into the *ompW*, *attB* in PP*ompW* (later renamed *attB*-in-PP_1) and *yecE* loci in 56 O145:H28 draft genomes was first examined by a BLASTN search as outlined in S10A Fig. The integration of Stx phages into the *attB* in PP*ompW* and/or *yecE* was determined by long PCR amplification using primers targeting the *stx* genes and sequences adjacent to these integration sites, as schematically shown in S10B Fig. The products of long PCR were

used for sequence determination of each prophage. The primers used in this analysis are listed in S7 Table.

Sequencing libraries were prepared for each product of long PCR (ranging from 15 to 33 kb) using the Nextera XT DNA Sample Preparation Kit (Illumina) and sequenced on the Illumina MiSeq platform to generate paired-end (PE) reads (300 bp x 2). Prophage genome sequences were obtained by assembling and scaffolding Illumina PE reads using the Platanus_B assembler (v1.1.0) (http://platanus.bio.titech.ac.jp/platanus-b) [36]; then, gaps were closed by Sanger sequencing PCR products that spanned the gaps. Annotation of all prophage genomes was carried out with DFAST, followed by manual curation using IMC-GE software (In Silico Biology). All sequences have been deposited in the DDBJ/EMBL/GenBank databases under the accession numbers listed in S1 Table. GenomeMatcher (v2.3) [37] was used for genome sequence comparison and to display the results.

### Searches for PP*ompW*s and the 21-bp *attB* sequences in the complete *E. coli* genomes

Serotypes and *stx* and *eae* subtypes of the 767 complete *E. coli* strains were determined by BLASTN as previously described [27,29]. Systematic ST determination was performed by a read mapping-based strategy using the SRST2 program [38] with default parameters. Read sequences of the complete genomes were simulated with the ART program (ART_Illumina, version 2.5.8) [39]. The genomes whose ST was not precisely defined (possible ST containing a novel allele, an uncertain ST, and no STs in the present database) were reanalyzed using MLST 2.0 with "Escherichia coli #1" schemes [40] (https://cge.cbs.dtu.dk/services/MLST/).

The presence of PP*ompW* and the 21-bp *attB* sequence in PP*ompW* was examined in the complete genomes by a BLASTN-based search as follows. The presence of PP*ompW* was determined using two query sequences: one was the integrase gene of O145:H28 strain 112648 (EC112648_1574) (thresholds: >90% identity and >90% coverage), and the other was the *ompW*-containing region on the chromosome of *E. coli* K-12 (No. NC_000913; nucleotide positions 1,314,020–1,315,224; no phage integration) to examine the absence of phage insertion into the *ompW* locus (threshold: >85% identity and <60% coverage). When either the integrase gene or the *ompW* locus split by some insertion was detected, we analyzed the gene organization of these regions to determine if PP*ompW* was present. The search for the 21-bp *attB* sequence (5'-GTCATGCAGTTAAAGTGGCGG-3') (S1C Fig) was performed with the blastn-short task option (thresholds: >95% identity and 100% coverage). The 21-bp sequences in the *yecE* gene, which were similar to the 21-bp *attB* sequence in PP*ompW*, were removed.

### SNP detection and phylogenetic analysis

The SNP sites (3,277 sites) of the core genomic sequences of the 64 O145:H28 strains were detected by MUMmer [41], followed by filtering recombinogenic SNPs by Gubbins [42], and used for reconstruction of an ML tree in RAxML [43] with the GTR gamma substitution model as previously described [27]. To reconstruct the phylogeny of the *E. coli* strains carrying PP*ompW*, we used 92 *E. coli* strains representing each of the 92 serotypes that contained PP*ompW*-carrying strains. Strains in which the 21-bp *attB* sequence was detected were preferentially selected from the serotypes that contained multiple strains. *Escherichia* cryptic clade I strain TW15838 (No. AEKA01000000) was used as an outgroup. The core genes (n = 2,642) of these strains, which were defined as the genes present in 100% of strains, were identified by Roary [44], and their concatenated sequence alignments were generated by the same software. Based on the alignment (109,927 SNP sites in total), an ML tree was constructed using RAxML as described above. Phylogroups of the strains were determined by ClermonTyping [45]. ML

trees were displayed and annotated using iTOL [46] or FigTree (v1.4.3) (http://tree.bio.ed.ac.
uk/software/figtree/).

## Supporting information

**S1 Fig. Determination of the *attP* sequences of phages integrated in *ompW*, PP*ompW*, and
*yecE* in O145:H28 strains 12E129 and 112648.** (A) Schematic representation of the PCR
strategy used to amplify the *attP*-flanking region (left panel) and the locations of PCR primers
used for each phage (right panel). (B) PCR detection of excised and circularized prophage
genomes. Total cellular DNA isolated from MMC-treated (+) or MMC-untreated (-) cells was
analyzed. A chromosome backbone (CB) region was amplified as a positive control. (C) The
*att* sequences of the three prophages in strain 12E129. The *attP*-containing sequences obtained
by sequencing the PCR products shown in S1B Fig were aligned with the *attR*-, *attL*-, *attB*-
containing sequences to define the *att* sequences of each phage. The *ompW* sequences of strain
K-12 MG1655 (accession No. NC_000913) and the *yecE* sequence of O145:H28 strain 122715
(accession No. AP019708), in which no phages were integrated, were used as the *attB*
sequences, respectively (indicated by a dagger and a section mark, respectively). The defined
*att* sequences are indicated by uppercase letters. (D) Detection of packaged DNA of the three
prophages in the DNase-treated lysates of strain 112648 with (+) or without (-) MMC treat-
ment. The CB region was amplified as a negative control. The same primer pairs as shown in
S1A Fig were used in this analysis.
(TIF)

**S2 Fig. All-to-all genome sequence comparison of PP*ompW*s and that of prophages in the
*attB* in PP*ompW* and PP*yecE*s found in 64 O145:H28 strains.** Dot plot matrixes of the
concatenated sequences of the 20 PP*ompW* genomes (A) and 27 prophage in the *attB* in
PP*ompW* and PP*yecE* genomes (B) found in 64 O145:H28 strains are shown. Strain names and
information on the ST and ST32 clade of each strain are indicated. Sequence identities are
indicated by a heatmap. In panel A, the nucleotide sequences between the *attB* in PP*ompW*
and the *attR* (approximately 428 bp in length) were excluded from this analysis because the
sequences of this region in three strains (499, EH1910, and KIH15-140) were not determined.
Average nucleotide identities (ANIs) among these genomes ranged from 97.0% to 99.9%. In
panel B, the subtype of Stx encoded by each prophage and the integration site of each phage
are indicated. Prophage groups sharing similar genomic sequences are framed by boxes. ANIs
among the Stx1a phages ranged from 98.1% to 99.9% and those between Stx2a phage of strain
RM9872 and the other Stx2a phages of clade A/B strains framed by purple boxes showed over
98.1%.
(TIF)

**S3 Fig. Sequence comparison of the two copies of Stx2 phage found in four *E. coli* genomes.**
Dot plot matrixes of the concatenated sequences of the Stx phages found in four strains (two
O145:H28 strains, an O157:H7 strain, and an O145:H25 strain) in their *attB* in PP*ompW* and
*yecE* loci are shown. The names of host strains, Stx2 subtypes, and integration sites of each
Stx2 phage are indicated. Stx2 phages in the same strain are framed by boxes. The two Stx2
phages in three strains (indicated by purple boxes) showed high sequence identity across their
entire genomes (ANI: >99.8%), suggesting that they were duplicated in each strain. Sequence
identities are indicated by a heatmap.
(TIF)

**S4 Fig. Comparison of the PP*ompW* genomes containing the 21-bp *attB* sequence.** In the
left panel, along with the same ML tree as shown in Fig 3, *E. coli* strain ID, phylogroup (PG),

and the presence (colored) or absence (open) of the 21-bp *attB* sequence in each *E. coli* are indicated. In the right panel, the genome structures of PP*ompW*s containing the *attB* are drawn to scale. In two strains indicated by asterisks (E473 and E471), recombination between PP*ompW* and another prophage along with translocation of chromosome segments caused complicated chromosome inversions around the replication terminal; therefore, only relevant prophage regions are shown. Homologous regions and sequence identities are depicted by shading with a color gradient. The Stx2a phages integrated into the *attB* locus in strains E474 and E118 are schematically indicated.
(TIF)

**S5 Fig. Variation in the prophage integration patterns in the prophage clusters that contained prophages carrying potential *attB* sites.** In the left table, a list of 33 strains that possessed prophage clusters that contained prophages carrying the 21-bp sequence identical or nearly identical to that of the *attB* in PP*ompW* is provided. In the right panel, the patterns of phage integration are schematically illustrated. Strains showing each pattern are also indicated in the left table. CDSs shown by colored triangles include pseudogenes. The 21-bp sequence (renamed *attB*-in-PP_1) and other *attB* sequences are indicated. Among these sequences, the two indicated by an asterisk are truncated by IS insertion. Several *attB* sequences are missing because of deletions. The T3SS effector set (light green triangles) consists of any of the seven effector family/subfamily genes that are encoded by the PP*ompW* EELs shown in Fig 3. Prophage that are apparently defective due to multiple gene degradation and deletion are indicated by (d). Genomic structures of four prophage clusters (indicated in bold in the left table) are presented in Fig 4. Types a, c, and d include a minor variation; homologous recombination between the second PP*ompW* and the first PP*ydfJ* (type a2), integrase-deficient PP*ydfJ*s with or without additional phage integration in tandem (types c2 or c3, respectively), and a region comprising two degraded prophages integrated in tandem between the *trg* and *rspB* genes without phage integration into the *attB*-in-PP_2 locus (type d2) are shown.
(TIF)

**S6 Fig. The *attB*-in-PP_2 sequences.** (A) Locations of the *attB*-in-PP_2 sequences in representative prophage genomes. (B) Comparison of the nucleotide sequence of *attB*-in-PP_2 among the prophages shown in panel A.
(TIF)

**S7 Fig. The *attP* sequence of PP*ydfJ*.** (A) Schematic representation of the *ydfJ*-flanking region and the prophage clusters present at the *ydfJ* locus in three *E. coli* strains. Because the integrase genes of the PP*ydfJ*s in strain PV15-279 (PP*ydfJ*-L and PP*ydfJ*-R) have both been inactivated by IS insertion, the PP*ydfJ*-R of O26:H11 strain 11368 was used for sequence determination of the *attP*-flanking region of PP*ydfJ* by sequencing a PCR amplicon obtained with two primers (indicated by red and blue arrows). (B) The *att* sequences of the four PP*ydfJ*s. The *attP*-containing sequence of the PP*ydfJ*-R of strain 11368 was aligned with the *attR*-, *attL*-, and *attB*-containing sequences to define the *att* sequences of each phage. Because phages are integrated in the *ydfJ* locus in many *E. coli* strains including K-12, the *ydfJ* sequence of O104:H4 strain C227-11, in which no phage was integrated in this locus, was used as the *attB* sequence. The 18- or 19-bp *att* sequence that we defined is indicated by uppercase letters.
(TIF)

**S8 Fig. The *attB*-in-PP_1 and its flanking sequences in prophages and comparison with the *E. coli yecE* sequence.** (A) The locations of the *attB*-in-PP_1 (initially called 21-bp *attB* in PP*ompW*) sequences in the genomes of six PP*ompW*s and three other phages integrated in prophages and in the *yecE* locus of *E. coli* O145:H28 strain 122715. The 21-bp *attB*-in-PP_1

sequence and the additional 79-bp sequence homologous to the *yecE* gene are indicated by red and purple, respectively. The *attB*-in-PP_2 and *attB*-in-PP_3 are also indicated by blue and orange, respectively. The sequences of the two regions indicated by green are conserved between prophages with up to 5 SNPs. The lengths of the two regions are 185 bp (left) and 228 bp (right). (B) Alignment of the 100-bp sequences homologous to the *yecE* locus in the nine prophages shown in panel A with the corresponding sequence of the *yecE* locus of strain *E. coli* O145:H28 strain 122715. The 21-bp *attB*-in-PP_1 sequence is indicated by uppercase letters. The 100-bp sequences of these prophages were 87% identical to the *yecE* sequence.
(TIF)

**S9 Fig. The *attB*-in-PP_2 sequence and its flanking sequences.** (A) The locations of the *attB*-in-PP_2 sequences (blue) in eight prophage genomes and on the chromosome of *E. coli* K-12 strain MG1655. The 96-bp *attB*-in-PP_2 sequences and their flanking sequences (184 bp and 29 bp in length) homologous to the *ykgJ-ecpE* region on the *E. coli* MG1655 chromosome are indicated by blue, pink, and dark brown, respectively. The presence of *stx* and T3SS effector genes in each prophage is also indicated. (B) Alignment of the *attB*-in-PP_2 and its flanking sequences in the prophages shown in panel A with the corresponding sequence of the *ykgJ-ecpE* region on the *E. coli* MG1655 chromosome. Only the prophage genomic regions homologous to the *ykgJ-ecpE* region are shown. The 184-bp regions (pink) of prophages show 83% sequence identity with the *ykgJ-ecpE* region. Note that the 96-bp *attB*-in-PP_2 (blue; indicated by uppercase letters) contained 23 SNPs.
(TIF)

**S10 Fig. Procedures to determine the phage integration into the *ompW*, 21-bp *attB* in PP*ompW* (later in the manuscript, renamed *attB*-in-PP_1) and *yecE* loci.** (A) Analysis of phage integration by a BLASTN search. Draft genomes of O145:H28 (n = 56) were searched by BLASTN, using the sequences of the *attL*- and *attR*-containing regions of the prophages at *ompW*, *attB* in PP*ompW* and *yecE* in strain 112648 (P08L/R, P09L/R, and P12L/R, respectively) as queries. Each query sequence was composed of the sequences from the host chromosome and prophage (60 bp each) with the *att* sequence determined in this study (121 bp for P08 and 21 bp for P09/P12) located between them. Phage integration at each locus was considered positive when *attL*- and *attR*-containing sequences were both detected (identity threshold: >95%). Phage integration in all but two genomes was determined by this analysis. In strains EH1910 and H27V05, although phages integrated into *yecE* (PP*yecE*) were detected, PP*ompW* was not detected. Unexpectedly, however, the P09L/R sequences (corresponding to the *attL*- and *attR*-containing sequences of the prophage in PP*ompW*) were detected in EH1910, and a partial P09 *attL* sequence (74.5% coverage) was detected in H27V05. Therefore, the *ompW* and *attB* in PP*ompW* loci of the two genomes were defined as 'Others', and subjected to long PCR analysis along with the identified prophages. (B) Long PCR analysis and sequence determination of prophage genomes. Strategies for five types of analysis are shown. Type I analysis: The genomes of PP*ompW*s that did not contain prophages were divided into three segments and amplified by three long PCRs to obtain the PCR products for genomic sequence determination. Note that the left and right segments included the left and right PP*ompW*-chromosome junctions, respectively (the same strategy was employed in Types II-V analyses). Type II analysis: The genomes of PP*ompW*s that contained an Stx phage were amplified together with the Stx phage genomes using 5 or 6 primer pairs to confirm the presence of these prophages and to obtain the PCR products for genome sequence determination. Two primers targeted the *stx* gene (*stx1* or *stx2*). As we detected recombination between the Stx phage and a prophage located at the *ydfJ* locus in two strains (EH1910 and 499), a different primer (the leftmost one) was used, thus labeled Type IIb. Type III analysis: In four strains, in

which the PP*ompW* contained an Stx phage, the genome of PP*ompW* and the early region of the Stx phage were amplified using 4 primer pairs, and only these genomic regions were sequenced. Type IV and V analyses: The genomes of PP*yecE*s were amplified using 2 or 3 primer pairs to obtain the PCR products for genomic sequence determination. When the PP*yecE* contained the *stx* gene (Type IV), two *stx*-targeting primers were used as in Type II analysis. For the PP*yecE* in strain H27V05 (Type Va), only the early region was amplified and sequenced.
(TIF)

**S1 Table. *E. coli* O145:H28 strains analyzed in this study.**
(XLSX)

**S2 Table. The proportion of STEC/EPEC strains in the complete *E. coli* genome set analyzed in this study (last access: 20th of July 2019).**
(XLSX)

**S3 Table. *E. coli* strains containing the 21-bp *attB* sequence found in prophage at non-*yecE* loci.**
(XLSX)

**S4 Table. *E. coli* O157:H7 and O145:H25 strains carrying prophages integrated in PP*ompW*.**
(XLSX)

**S5 Table. All available complete *E. coli* genomes in the NCBI database (last access: 20th of July 2019).**
(XLSX)

**S6 Table. Primers used for PCR amplification for prophage regions.**
(XLSX)

**S7 Table. Primers used for long PCR analysis.**
(XLSX)

## Acknowledgments

We thank M. Horiguchi, M. Kumagai, Y. Nagayoshi, and K. Ozaki for providing technical assistance. We also thank Prof. J.G. Mainil, Prof. D. Piérard, and the EHEC working group in Japan for providing O145:H28 strains.

## Author Contributions

**Conceptualization:** Keiji Nakamura, Tetsuya Hayashi.

**Data curation:** Keiji Nakamura.

**Formal analysis:** Keiji Nakamura, Yoshitoshi Ogura, Yasuhiro Gotoh.

**Funding acquisition:** Keiji Nakamura, Tetsuya Hayashi.

**Investigation:** Keiji Nakamura.

**Methodology:** Keiji Nakamura, Yoshitoshi Ogura, Yasuhiro Gotoh.

**Project administration:** Tetsuya Hayashi.

**Resources:** Keiji Nakamura, Yoshitoshi Ogura.

**Visualization:** Keiji Nakamura.

**Writing – original draft:** Keiji Nakamura.

**Writing – review & editing:** Yoshitoshi Ogura, Yasuhiro Gotoh, Tetsuya Hayashi.

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
