## [Decision Letter · Decision Letter 0]

21 Dec 2020

Dear Pr Tetsuya,

Thank you very much for submitting your manuscript "Prophages integrating into prophages: a mechanism to accumulate type III secretion effector genes and duplicate Shiga toxin-encoding prophages in Escherichia coli" for consideration at PLOS Pathogens. As with all papers reviewed by the journal, your manuscript was reviewed by members of the editorial board and by several independent reviewers. In light of the reviews (below this email), we would like to invite the resubmission of a significantly-revised version that takes into account the reviewers' comments.

The manuscript has been reviewed by three reviewers as well as myself. They all feel as though there is a significant amount of very well performed work in the study, there are a number of things that could be improved in the manuscript to result in a stronger, focuced and more comprehensive manuscript.

The key points for the authors to review/address in a resubmission would be:

1) the inclusion of some detail about the selection of the isolates to be included in this study.

2) further clarification of the inclusion and impact of the T3SS effectors. This is currently unclear in the current version of the manuscript and is not well supported in all of the serotypes that were examined. It may be helpful to limit the analysis to a few serotypes to clarify the narrative.

We cannot make any decision about publication until we have seen the revised manuscript and your response to the reviewers' comments. Your revised manuscript is also likely to be sent to reviewers for further evaluation.

Sincerely,

David A Rasko

Guest Editor

PLOS Pathogens

Christoph Tang

Section Editor

PLOS Pathogens

Kasturi Haldar

Editor-in-Chief

PLOS Pathogens

orcid.org/0000-0001-5065-158X

Michael Malim

Editor-in-Chief

PLOS Pathogens

orcid.org/0000-0002-7699-2064

The manuscript has been reviewed by three reviewers as well as myself. They all feel as though there is a significant amount of very well performed work in the study, there are a number of things that could be improved in the manuscript to result in a stronger, focuced and more comprehensive manuscript.

The key points for the authors to review/address in a resubmission would be:

1) the inclusion of some detail about the selection of the isolates to be included in this study.

2) further clarification of the inclusion and impact of the T3SS effectors. This is currently unclear in the current version of the manuscript and is not well supported in all of the serotypes that were examined. It may be helpful to limit the analysis to a few serotypes to clarify the narrative.

Reviewer's Responses to Questions

**Part I - Summary**

Reviewer #1: Nakamura et al report an impressive, elegant and very precise analysis of a new observation, namely the faculty for a prophage to host another prophage inside its genome, which they name PP in PP for short. They detail this observation starting from a set of STEC strains of E. coli, but then extend this analysis across all E. coli, and end up suggesting that it may be an even more frequent phenomenon in nature.

Reviewer #2: Compound prophage regions are a feature of some STEC genomes especially near the terminus of O157 strains. The prophages that are present and recombination between them underpins the majority of structural variation in these strains. New incoming phages with lysogenic potential can only integrate if there is an available attB (integration site in the bacterial xome) and certainly for the majority of Shiga toxin-encoding prophages which are generally Lambda-like, then these sites have been very well characterized including wrbA, yehV, sbcB and yecE.

The manuscript by Nakamura et al is primarily descriptive and based on genome analysis initially of STEC O145:H28 isolates followed by casting the net wider to other STEC and more general E. coli. Its key observation is that Stx2a-encoding and other prophages can insert into an ‘ompW’ attB site near the end of a prophage already inserted into the ‘original’ ompW-attB site. In this way they state that compound phages are produced by integration of one phage into another. They surmise that the PP-attB site is the likely integration site by showing that the Stx2a prophages can excise cleanly from this site. An important observation of this study though is that the original PP-ompW is fixed and does not excise as a functional phage, perhaps not surprisingly given the integrated phage within it.

The study demonstrates that there is a more or less conserved prophage at this site in this serotype and that has this prophage attB site and so can presumably receive prophages. The critical thing from my point of view for the overall significance of the manuscript is how general is such acquisition, and therefore compound phages built this way, and the process by which an integrated prophage would ‘acquire’ such an attB site. As suggested by the authors the most likely is duplication and recombination but this process is unknown at this point. Intra-chromosomal recombination events are common in these strains, enabled by multiple similar prophage regions as well as IS elements that promote homologous recombination. From one perspective it is not surprising that an attB site could be duplicated as part of a large duplication potentially with inversion and deletion events. It is also possible that the Stx2a-encoding phage is actually copied as part of this process without necessarily being inserted as a separate event. A ‘chicken and egg, argument. Linked to this I think they are missing a trick by not including a wider chromosome structure to prove its phages moving from yecE to the att inside the phage at ompW and while less likely that it’s a large structural variant this is a possibility as the prophages at ompW/RspR/mlrA are almost always compounded because of large-scale genome recombination events.

Reviewer #3: Strengths:

Analysing prophage regions of STEC genomes is very challenging and this study provides an approach on how to do that.

The manuscript is very well written.

The analysis is complex but the results section is well structured and easy to follow.

The analysis is novel and the extent to which prophage-prophage integration occurs is a significant finding.

The paper describes a mechanism by which E. coli can accumulate virulence determinants thus providing insight into how highly pathogenic variants of STEC evolve.

Understanding mechanisms of loss and acquisition of stx-encoding prophage is key to enabling us to monitoring for emerging threats to public health.

Weaknesses:

The authors should explain their strain selection criteria.

A wide range of STEC serotypes have been selected for the study - the advantage of this is that it shows this phenomenon is widespread but it may be difficult for readers unfamiliar with STEC to assimilate the data.

I think a clear explanation as to why this specific strain set was selected for this study would help.

**Part II – Major Issues: Key Experiments Required for Acceptance**

Reviewer #1: I have very little questions/suggestions to improve this already very complete work.

1. From Figure 1c one gathers that the lambda in ompW is defective in strain 12E129. It has indeed an IS in the structural region. Do you have elements to suggest that this is always the case when the PPompW prophage is hosting a PP-in-PP? Do they all have IS for instance? More broadly, are all PPompW predicted defective, regardless of their “occupation” (44% of E. coli strains having PPompW seems a lot…)?

2. The fact that strain 12E129 hosts two copies of prophage Stx2a suggests that immunity to super-infection did not “work”. Or was it a single infection followed by two simultaneous integration events? Are the two copies in direct or inverse orientation on the chromosome of E. coli?

3. From Fig2, it seems that the PP-in-PP event occurs only when the first att site (yecE) is occupied. Why should it be so?

Reviewer #2: The authors analyse very different serotypes in the end, which I understand are there to see how broad the concept they are proposing might be. But the way it is presented actually makes the work harder to follow as the sites described soon spiral out to related putative attB sequences with no functional analysis.

The work is much tighter and easier to follow focused on O145. Conceptually, active phage cannot really have their own attB site within them as the moment they integrate they would become a target for prophage insertion and so would ‘kill themselves off’, so the event here is one outcome of chromosomal rearrangements involving prophage that are important for generating strain diversity.

While the compound phage origins and associated genomics are intriguing and coherent, I am less convinced by the argument then used in relation to acquisition of putative T3SS effectors. This seems to be a focus more to fit in with a pathogenesis argument when the attB PP in PP concept which is more about general phage evolution. Yes, STEC strains with a LEE locus have acquired effectors over time, the majority encoded on prophages. The additional attB site does allow for an extra inserted/duplicated phage but there is no evidence provided that this organization can facilitate linked effector expression or if there is any other advantage over integration at any other site. I also found it difficult to work out from the supplementary data if the effectors were restricted to eae+ strains, i.e. were only/mainly present in strains with a T3SS.

I think they have shown that such attB duplication/evolution allows more prophage based information to be held in the genome but the origin of such sequences and any advantage in relation to the T3S effector profile for any specific strain/serotype is not supported experimentally, so there is a fair amount of speculation.

Reviewer #3: None identified

**Part III – Minor Issues: Editorial and Data Presentation Modifications**

Reviewer #1: Minor points

4. Line 67: ”locus-encoding”, remove locus-

5. Line 72: it was hard at the onset to understand precisely what the duplication meant. I propose some rewriting such as :”we initially analyzed the two copies of an Stx2-encoding PP present in two loci of a STEC O145:H128 genome, …”

6. Lines 84-88: along the same line, to help the reader enter into the PP jungle, some suggestions: “As the lambda-like P08 prophage was also found at ompW”. The next sentence begining with “However, by analyzing” is really hard to follow, and important for the following. Avoid the “respectively” formula and may be, cut the sentence. This onset of the work has to be crystal clear. I had to draw a little cartoon for myself to make it clear.

7. Line 152: "by filtering the att sequence in yecE", you mean you masked it?

8. Line 162: I am a bit lost here, about the 145 strains found in 4 different phylotypes: do you mean that these strains all belong to serotype O145:H28 except 3?

Reviewer #2: Really well produced supplementary figures, especially Fig. S4. An incredible amount of detail about the possible construction of certain composites at specific sites, but not sure about the link to the effector and impact/role on virulence or regulation.

All phage comparisons are done by BLAST and are only in text saying they are similar or not, I think it would be nice to back this up with some numbers i.e. % similarity of prophage gene content.

If they could show a mechanism behind such attB duplication or transposition to argue that this was an evolved general strategy to acquire more exogenous prophage information then that would be exciting. Another issue is that such PP expansion cannot continue as it will imbalance the replichores unless centered close to the terminus (as in this case?) or matched by reciprocal events on the other side of the chromosome?

I find their nomenclature of ompW-PP and att-in-ompW-PP and stx2a-PP does make it harder to read rather than easier.

Reviewer #3: As stated above:

The authors should explain their strain selection criteria.

A wide range of STEC serotypes have been selected for the study - the advantage of this is that it shows this phenomenon is widespread but it may be difficult for readers unfamiliar with STEC to assimilate the data.

I think a clear explanation as to why this specific strain set was selected for this study would help.

PLOS authors have the option to publish the peer review history of their article (what does this mean?). If published, this will include your full peer review and any attached files.

Reviewer #1: No

Reviewer #2: No

Reviewer #3: No
---

## [Decision Letter · Decision Letter 1]

14 Apr 2021

Dear Pr Tetsuya,

We are pleased to inform you that your manuscript 'Prophages integrating into prophages: a mechanism to accumulate type III secretion effector genes and duplicate Shiga toxin-encoding prophages in Escherichia coli' has been provisionally accepted for publication in PLOS Pathogens.

Best regards,

David A Rasko

Guest Editor

PLOS Pathogens

Christoph Tang

Section Editor

PLOS Pathogens

Kasturi Haldar

Editor-in-Chief

PLOS Pathogens

orcid.org/0000-0001-5065-158X

Michael Malim

Editor-in-Chief

PLOS Pathogens

orcid.org/0000-0002-7699-2064

Reviewer Comments (if any, and for reference):

Reviewer's Responses to Questions

**Part I - Summary**

Reviewer #1: My remarks have been answered appropriately. The paper is now easier to read, it is a good thing to have limited the occurences of the “PP” abbreviation, it seems however that the first occurrence, line 87, has not been removed.

Reviewer #2: I thank the authors for trying to address some of the points I have raised. I still feel that the argument around a general mechanism to acquire multiple type 3 secretion-related effectors is not that well supported. In the revised Table S5, it is clear that there is one major prophage with a specific set of effectors: nleG1/nleA/nleH/nleF/nleG2/espM/nleG3 that has integrated at this site across multiple backgrounds, although I agree there are a few alternative versions. My take home is that a prophage has become cryptic and as part of that process, or later, it has acquired an attB site, and then other specific prophage can be integrated and selected if they work in that background. I don't think this is about active prophage with extra attB sites, i.e. they have to be cryptic, so its more about prophage sites that are more likely to be sites of recombination and in this case an extra attB site has been generated. It is really well presented and lovely work but from my perspective it has been packaged well to promote this single point which is (IMHO) a predictable result/example of what such general prophage-based recombination can generate.

**Part II – Major Issues: Key Experiments Required for Acceptance**

Reviewer #1: (No Response)

Reviewer #2: (No Response)

**Part III – Minor Issues: Editorial and Data Presentation Modifications**

Reviewer #1: (No Response)

Reviewer #2: (No Response)

PLOS authors have the option to publish the peer review history of their article (what does this mean?). If published, this will include your full peer review and any attached files.

Reviewer #1: No

Reviewer #2: No

---

## [Editor Report · Acceptance letter]

26 Apr 2021

Dear Pr Tetsuya,

We are delighted to inform you that your manuscript, "Prophages integrating into prophages: a mechanism to accumulate type III secretion effector genes and duplicate Shiga toxin-encoding prophages in Escherichia coli," has been formally accepted for publication in PLOS Pathogens.

Best regards,

Kasturi Haldar

Editor-in-Chief

PLOS Pathogens

orcid.org/0000-0001-5065-158X

Michael Malim

Editor-in-Chief

PLOS Pathogens

orcid.org/0000-0002-7699-2064